

# A pan-cancer analysis of prognostic genes

Jordan Anaya[1,2], Brian Reon[1], Wei-Min Chen[3,4], Stefan Bekiranov[1] and Anindya Dutta[1]

[1] Department of Biochemistry and Molecular Genetics, University of Virginia, Charlottesville, VA, United States
[2] omnesres.com, Charlottesville, United States
[3] Center for Public Health Genomics, University of Virginia, Charlottesville, VA, United States
[4] Department of Public Health Sciences, Biostatistics Section, University of Virginia, Charlottesville, VA, United States

## ABSTRACT

Numerous studies have identified prognostic genes in individual cancers, but a thorough pan-cancer analysis has not been performed. In addition, previous studies have mostly used microarray data instead of RNA-SEQ, and have not published comprehensive lists of associations with survival. Using recently available RNA-SEQ and clinical data from The Cancer Genome Atlas for 6,495 patients, we have investigated every annotated and expressed gene's association with survival across 16 cancer types. The most statistically significant harmful and protective genes were not shared across cancers, but were enriched in distinct gene sets which were shared across certain groups of cancers. These groups of cancers were independently recapitulated by both unsupervised clustering of Cox coefficients (a measure of association with survival) for individual genes, and for gene programs. This analysis has revealed unappreciated commonalities among cancers which may provide insights into cancer pathogenesis and rationales for co-opting treatments between cancers.

## INTRODUCTION

Led by The Cancer Genome Atlas, unprecedented efforts have been made to understand the molecular basis of cancer (http://cancergenome.nih.gov). Using standardized procedures, the TCGA Research Network has used whole genome sequencing, exome sequencing, RNA-SEQ, small RNA-SEQ, bisulfite-SEQ, and reverse phase arrays to identify the pathways commonly altered in different cancers (*Brennan et al., 2013*; *Cancer Genome Atlas N, 2012a*; *Cancer Genome Atlas, 2012b*; *Cancer Genome Atlas Research, 2011*; *Cancer Genome Atlas Research, 2012*; *Cancer Genome Atlas Research, 2013a*; *Cancer Genome Atlas Research, 2013b*; *Cancer Genome Atlas Research, 2014a*; *Cancer Genome Atlas Research, 2014b*; *Cancer Genome Atlas Research, 2014c*; *Cancer Genome Atlas Research, 2014d*; *Cancer Genome Atlas Research et al., 2013a*). As a result, we now know the most commonly mutated genes in dozens of cancers and can use this information to give patients targeted therapeutics.

Whereas well established statistical techniques exist for identifying mutations which are drivers instead of simply passengers (mut-drivers), identifying copy number aberrations,

Corresponding authors
Jordan Anaya,
omnesresnetwork@gmail.com
Anindya Dutta, ad8q@virginia.edu

methylation changes, or non-coding mutations that alter expression of a gene and result in a growth advantage (epi-drivers) are more difficult to identify and represent a "dark matter" of cancer (*Vogelstein et al., 2013*). Although it currently is challenging to identify epi-drivers which lead to development of a cancer (tumorigenesis), by correlating these changes to survival it is possible to detect their role in disease progression (pathogenesis), which is one of main goals of cancer research.

Of the possible genomic measures that can be correlated with survival, gene expression has been shown to be the strongest predictor of survival (*Zhao et al., 2015*), which is intuitive given that gene levels together with protein levels and posttranslational modifications are the final readout of the different possible alterations in a cell and are the final effectors of phenotype. To date many attempts have been made to identify genes whose expression is associated with survival to either identify markers that can predict patient survival or to identify mechanisms of pathogenesis (*Chen et al., 2007*; *Valk et al., 2004*; *Van de Vijver et al., 2002*). One of the success stories of this approach is the identification of HER2 in breast cancer patients and the development of herceptin (*Bange, Zwick & Ullrich, 2001*). This story also highlights the complications treatment regimens can have on interpreting survival data. Whereas HER2 overexpression used to predict poor survival for breast cancer patients, because of the progress of personalized medicine these patients now do well and HER2 would not show up as a prognostic marker in a data set with HER2 positive patients on herceptin. While treatments may introduce a confounding variable in understanding a disease, the ultimate goal of cancer studies is to improve patient outcome, and adding treatment to the equation adds more information and provides an opportunity to study genes in the context of the current standard of care.

The vast majority of studies to identify prognostic genes have focused on a single disease and have utilized microarrays instead of RNA-SEQ. In addition, these studies often only publish a small set of genes that together most significantly stratify patients. Even the TCGA Research Network publications do not provide lists of genes associated with survival. cBioPortal does allow users to make Kaplan-Meier plots for most of the cancers which contain survival information, but users still have to input one gene at a time, leaving one to wonder where researchers should go to find the genes which are most highly correlated with survival for their disease of interest.

Through the TCGA Network, RNA-SEQ has only very recently become available for thousands of human cancer samples. RNA-SEQ has multiple advantages over microarray data, including having a higher dynamic range, no probe affinity effects, ability to identify novel transcripts, and lower and consistently falling cost. We took advantage of the availability of this data to (1) investigate the ability of RNA-SEQ to associate expression with clinical outcome in a range of cancers, (2) perform the largest analysis of prognostic genes to date, and (3) provide every gene's correlation with survival for hypothesis testing and further investigations by the scientific community. In addition, attempts are now being made to identify commonalities between cancers with the hope that this type of analysis may be able to identify treatments that can be co-opted for a molecularly similar cancer. Given that only survival correlations integrate treatment with the genomic data,
prognostic genes represent an exclusive window for understanding how different cancers in the context of their individualized treatments relate to one another. By performing the largest pan-cancer analysis of prognostic genes, we identified groupings of cancers which were robustly replicated through several methods. This study serves as a starting point for better understanding how survival data can be used to understand the commonalities and differences of cancers.

## MATERIALS AND METHODS

### Code and files

All of the Python and R code used to generate the figures and tables in this study, including intermediate and final files, tables, and figures, is available at https://github.com/OmnesRes/pan_cancer. All scripts were run on a HP dv7t laptop with an i7-3820QM processor and 16GB of RAM running Windows 7, Python 2.7.5, and R 3.0.1.

### Construction of multivariate Cox models

RNA-SEQ and clinical data were downloaded from the TCGA data portal, https://tcga-data.nci.nih.gov/tcga/, in February 2015. For each cancer, survival information was parsed from the "clinical_follow_up" files and "clinical_patient" file, and for each patient the most recent follow up information found in the multiple files was kept. Sex, age, and histological grade data were extracted from the "clinical_patient" file. For each cancer, only patients that had a follow up time greater than 0 days and had complete clinical information were included in the model. TCGA has used two different methods of reporting expression values, RSEM and RPKM. RPKM is simply the reads per kilobase per million mapped reads, while RSEM is a normalized value outputted by the RSEM software (*Li & Dewey, 2011*). For each cancer, only genes which had a median RSEM value greater than 1 (for RNASeqV2), or median RPKM value greater than .1 (for RNASeq), and had 0 expression in less than one fourth of patients were included in the analysis. RNASeq uses a different gene annotation file from RNASeqV2, and because RNASeqV2 represents the most recent analysis, for RNASeq analyses only those genes present in the RNASeqV2 gene annotation file were included. Multivariate Cox models were run with the coxph function from the R survival library, and the equation for each model is shown in Table S1. Grade information was included in the model by separate terms, which were either 1 or 0, and model input gene expression values were inverse normal transformed. If a patient had replicates for their primary tumor, those expression values were averaged prior to inverse normal transformation. The scripts for performing Cox regression for each cancer are named "cox_regression.py."

### Gene set analysis

For each cancer, the 250 most significant protective genes and 250 harmful genes were inputted separately into MSigDB with the "positional genes sets," "chemical and genetic perturbations," "canonical pathways," "KEGG gene sets," "microRNA targets," "transcription factor targets," "cancer modules," "GO biological process," and "oncogenic signatures" sets selected: http://www.broadinstitute.org/gsea/msigdb. The FDR *q*-value

threshold was set at .05 and the top 100 enriched gene sets were saved, except for the 250 protective genes in BLCA, which only contained 27 overlaps below .05.

## Normalization of Cox coefficients

In order to compare the Cox coefficients between cancers we robustly scaled the negative and positive coefficients, $x$, to their 5th and 95th percentile values, respectively, using the following sigmoidal normalization function:

$$z_{\pm} = \frac{2}{1 + e^{-\frac{2x}{|u_{\pm}|}}} - 1$$

where $u_-$ and $u_+$ are the 5th and 95th percentile values of the negative and positive Cox coefficients, respectively. The implementation of this code is present in the files named "normalizing_coeffs.py," and all the normalized coefficients are listed in Table S1.

## Construction of gene programs

Gene programs from Table S4 of *Hoadley et al. (2014)* were used. In general, a nonredundant set of genes from gene sets which had a Pearson correlation of at least .9 (*Hoadley et al., 2014*) was generated for each program. The exact gene sets used are listed in Table S3. Lists of genes for the gene sets were obtained from http://www.broadinstitute.org/gsea/msigdb and *Fan et al. (2011)*.

## Determination of cancer subtypes

For Fig. 1B the subtype information was obtained from *Brat et al. (2015)* for LGG, and from *Cancer Genome Atlas Research (2014a)* for STAD. If subtype information for a patient was not available, the subtype was listed as "NA" in Fig. 1B.

# RESULTS

## Cancers vary in number of prognostic genes

In order to perform the most comprehensive cancer analysis possible, we selected TCGA cancers that had sufficient numbers of patients with RNA-SEQ data and mature clinical follow up information, and did not contain any publication restrictions. This resulted in us studying a total of 16 cancers, 10 of which were present in the original pan-cancer initiative (*Cancer Genome Atlas Research et al., 2013b*): acute myeloid leukemia (LAML), bladder urothelial carcinoma (BLCA), breast invasive carcinoma (BRCA), colon adenocarcinoma (COAD), glioblastoma multiforme (GBM), head and neck squamous cell carcinoma (HNSC), kidney renal clear cell carcinoma (KIRC), lung adenocarcinoma (LUAD), lung squamous cell carcinoma (LUSC), and ovarian serous cystadenocarcinoma (OV), and 6 cancers which have been the focus of limited individual or pan-cancer studies: cervical squamous cell carcinoma and endocervical adenocarcinoma (CESC), brain lower grade glioma (LGG), kidney renal papillary cell carcinoma (KIRP), liver hepatocellular carcinoma (LIHC), skin cutaneous melanoma (SKCM), and stomach adenocarcinoma (STAD).

We were interested in the effect a gene has on prognosis independent of factors such as tumor grade and age of a patient. To achieve this we used a multivariate Cox proportions hazards model (*Cox, 1972*), which is a standard regression method for studying survival data (*Claus et al., 2015*; *Gyorffy et al., 2013*; *Wu & Stein, 2012*; *Zhang et al., 2013*). For every cancer, a model was generated separately for each gene, with the number of covariates depending on the cancer. Unlike microarray data, RNA-SEQ data has extreme values which may affect regression. To account for this we inverse normal transformed the expression values of each gene, which has been shown to increase the sensitivity and specificity for multivariate regression with RNA-SEQ data (*Zwiener, Frisch & Binder, 2014*). Age and sex are also included in every model, and when a cancer contained strong histological grade information, grade was also included. If a patient was missing any of this information they were excluded from the analysis, and only primary tumors were considered, with the exception of SKCM, where metastatic tumors make up a large proportion of the patients.

A Cox model provides a *p*-value for each term in the model, indicating the significance of its association with the clinical outcome, and we recorded the *p*-values for every gene analyzed for the 16 different cancers. As can be seen in Table 1, there is a wide distribution among cancers in the number of genes that reached a Benjamini–Hochberg False Discovery Rate (FDR) adjusted *p*-value of less than or equal to .05. This can also be seen by looking at a distribution of the raw *p*-values for the different cancers (Fig. 1A and Fig. S1). This has important implications for understanding the significance of a gene being associated with survival in a specific cancer. For example, selecting a gene at random a researcher studying LGG has a 50% chance of being able to claim the gene is associated with survival, while a researcher studying STAD only has an 8% chance (using raw *p*-values).

Two factors that are known to be associated with power of a Cox model are sample size and number of events (deaths); however, looking through Table 1 it is difficult to find a pattern that can explain why certain cancers have more significant genes than other cancers. For example, BRCA has around twice the number of patients of any other cancer, but only has 30 genes that meet a FDR cutoff, which can be considered expressionally prognostic genes (EPGs). In contrast, KIRP has a fourth the number of patients of BRCA but has 2,415 EPGs. In addition, LUAD and LUSC have similar numbers of patients, median survivals, and events, yet have a large difference in EPGs. Interestingly, it has been shown that the number of prognostic genes for a cancer can be significantly different depending on whether microarray data or RNA-SEQ data is used (*Yang et al., 2014*). It is possible that the different numbers of EPGs between cancers are due to intra-disease heterogeneity and/or treatment differences that are not accounted for in the Cox model and are acting as confounding variables, or differences in the amount of transcriptional dysregulation between cancers.

## Protective and harmful genes display opposite expression patterns

The Cox model also provides a coefficient for each term, which is related to its contribution to the hazard ratio. A positive coefficient indicates that the gene increases the hazard ratio,

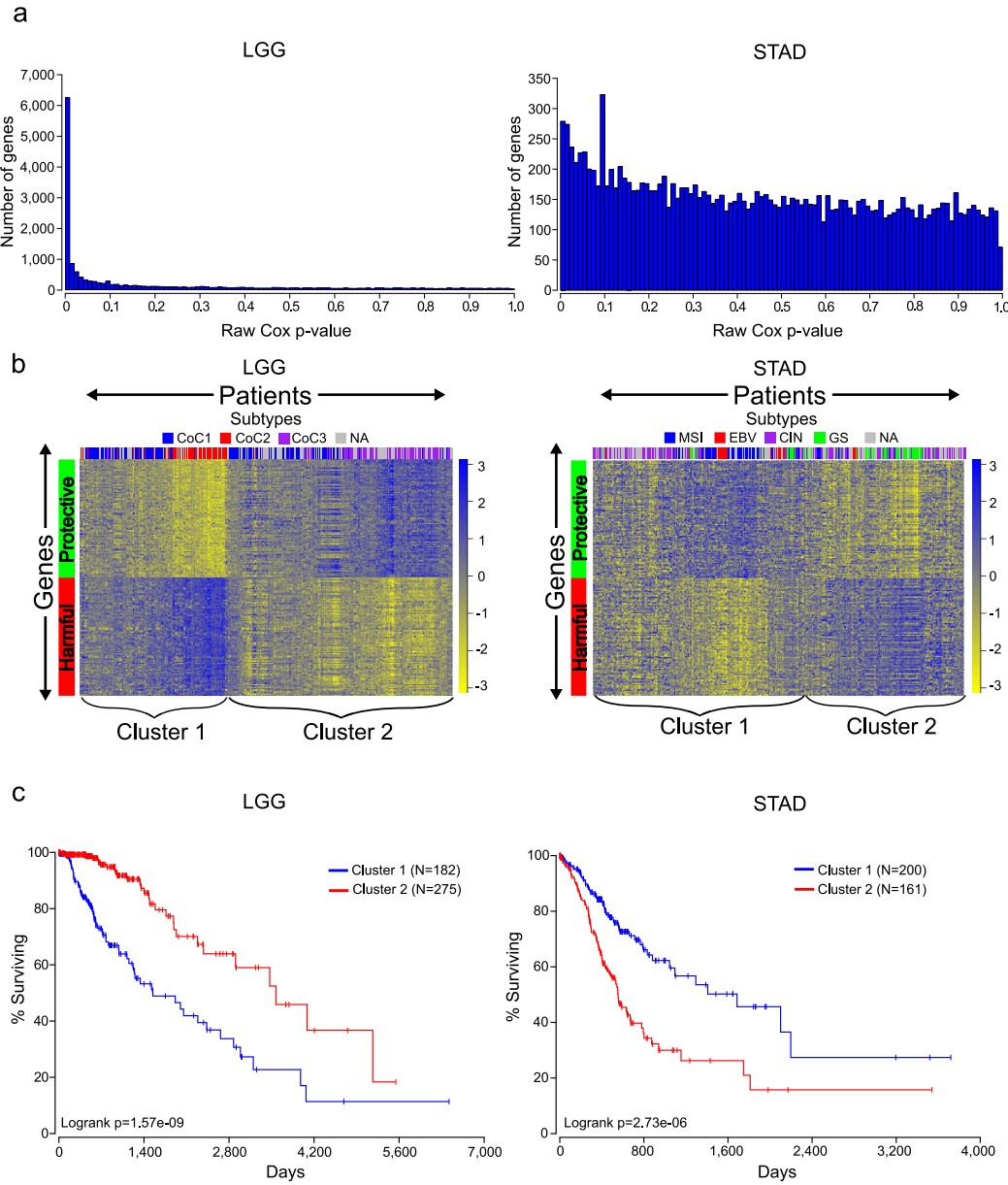

**Figure 1 Distinct expression patterns of protective and harmful prognostic genes.** (A) Raw gene *p*-value distributions from multivariate Cox models for a cancer with high number of expressionally prognostic genes (EPGs; LGG), and a cancer with low number of EPGs (STAD). Distributions for the other 14 cancers are displayed in Fig. S1. (B) Unsupervised hierarchical clustering (Pearson correlation distance metric) of patients using the inverse normal transformed expression values from the 100 most significant protective genes and 100 most significant harmful gene for LGG and STAD. (C) Kaplan-Meier plots comparing survival times for the two broad clusters of patients identified in (B) and logrank *p*-values for LGG and STAD.

**Table 1 Characteristics of datasets and patients included in this study.** Events are the number of deaths in the data set. Age is the average age and is in years. Median survival is in days. The median survival for KIRP could not be calculated.

| Cancer | Patients | Median survival | Events | Age at diagnosis | Male/female | RNASeqV2 | Genes in study | FDR ≤ .05 |
|--------|----------|-----------------|--------|------------------|-------------|----------|----------------|-----------|
| BLCA | 347 | 1,064 | 139 | 67.9 | 254/93 | YES | 16,385 | 532 |
| BRCA | 981 | 3,669 | 116 | 58.4 | 10/971 | YES | 16,649 | 30 |
| CESC | 259 | 3,097 | 60 | 48.1 | 0/259 | YES | 16,358 | 146 |
| COAD | 434 | 2,475 | 89 | 66.6 | 232/202 | YES | 16,414 | 0 |
| GBM | 152 | 406 | 119 | 59.8 | 98/54 | YES | 16,833 | 0 |
| HNSC | 484 | 1,671 | 190 | 61.2 | 353/131 | YES | 16,652 | 45 |
| KIRC | 516 | 2,386 | 167 | 60.7 | 335/181 | YES | 16,677 | 5,785 |
| KIRP | 247 | NA | 32 | 60.8 | 181/66 | YES | 16,430 | 2,415 |
| LAML | 149 | 577 | 92 | 54.7 | 80/69 | YES | 15,255 | 4 |
| LGG | 457 | 2,875 | 91 | 43.1 | 255/202 | YES | 16,818 | 7,186 |
| LIHC | 324 | 2,116 | 105 | 60.1 | 217/107 | YES | 15,855 | 2 |
| LUAD | 486 | 1,379 | 146 | 65.3 | 224/262 | YES | 16,784 | 1,179 |
| LUSC | 471 | 1,655 | 180 | 67.4 | 352/119 | YES | 16,979 | 0 |
| SKCM | 427 | 2,889 | 195 | 57.5 | 264/163 | YES | 16,067 | 1,548 |
| OV | 401 | 1,321 | 224 | 59.6 | 0/401 | NO | 15,748 | 0 |
| STAD | 361 | 874 | 130 | 65.2 | 238/123 | NO | 15,560 | 0 |

i.e., high expression of the gene correlates with earlier patient death, while a negative coefficient indicates that expression of the gene is protective. Using the cancer with the highest number of EPGs (LGG), we clustered patients with the 100 most significant genes which were harmful and the 100 most significant genes which were protective, and this revealed two broad clusters of patients: (1) those with high expression of harmful genes and low expression of protective genes, and (2) those with high expression of protective genes and low expression of harmful genes (Fig. 1B). Not surprisingly, performing a Kaplan-Meier analysis with these two groups revealed that cluster 2 has a much higher survival than cluster 1 (Fig. 1C). This result has important implications for trying to find gene sets which can most accurately predict patient survival. The similar expression patterns indicate that there are numerous combinations of genes that would only differ slightly in their ability to predict survival, making the identification of a 'best' set of genes somewhat meaningless. In addition, given that each gene individually had a $p$-value less than or equal to 1.4E−8, it is unlikely these patterns are due to chance but rather are the result of some underlying gene regulation and these genes may be members of known pathways.

Unlike LGG, some of the other cancers in our analysis yielded a much lower number of EPGs. While it might be tempting to disregard the results in these cancers, we decided to check if we could observe patterns of expression in the most significant good and bad genes like we had observed for cancers with a high number of EPGs. Clustering of the patients of STAD, which has one of the lowest numbers of EPGs, with the 100 most significant harmful genes, and the 100 most significant protective genes, again divided patients into two broad clusters. Surprisingly, a Kaplan-Meier analysis on these two groups showed a

very significant difference with a *p*-value of 2.73E-6 (Fig. 1C). This indicates that despite the fact that none of the genes in STAD meet a 5% FDR cutoff, they still contain important biological information. As a result, we decided to include all cancers in further analyses regardless of their numbers of EPGs.

## Cancers do not share prognostic genes, but do share gene sets

We were interested in seeing if the most significantly prognostic genes were shared across cancers. However, looking at the overlap of the 100 most significant genes across the 16 cancers revealed that there is very little overlap among the 16 cancers, consistent with previous results obtained from an analysis of four cancers (*Yang et al., 2014*) (Fig. 2A). Given the apparent co-regulation of the most significant genes in each cancer, we reasoned that although individual genes were not shared, maybe the genes were a part of gene sets which were shared between cancers. In addition, given that the harmful genes had an opposite pattern of expression from the protective genes, we hypothesized that they are regulated differently and would be enriched in different gene sets. To investigate this, we took the 250 most significant harmful genes and 250 most significant protective genes in each cancer, and separately found the 100 most enriched gene sets through MSigDB (*Subramanian et al., 2005*). Consistent with our idea that harmful and protective genes are regulated differently, there was very little overlap within a cancer between the 100 gene sets found with 250 harmful genes and the 100 gene sets found with 250 protective genes (Fig. 2B). In addition, the fact that even the protective and harmful gene sets from cancers with a low number of EPGs show almost no overlap reinforces the idea that prognostic genes in these cancers still contain biologically significant information.

Next we assessed the extent to which these protective and harmful gene sets overlapped between the different cancers. We investigated the overlaps separately for the 100 harmful gene sets and 100 protective gene sets (Figs. 2C and 2D). Overall there was more overlap between the harmful gene sets, and there were three cancers which clearly shared a high number of harmful gene sets, LUAD, LIHC, and KIRP. Investigating these overlaps further showed that the three cancers shared 58 gene sets, and LUAD and KIRP shared 85 gene sets (Fig. 2E). Looking at the overlaps of the protective gene sets, the largest overlap was between COAD and LUSC, and these cancers also shared gene sets with GBM (Fig. 2F).

We next asked what are the most common harmful and protective gene sets across cancers. Table S2 shows frequency of every gene set, with gene sets that were shared between harmful and protective sets within a single cancer marked in bold as they may be nonspecific. As might be expected, the most common gene sets observed for harmful genes were associated with poor differentiation and metastasis. In contrast, the protective gene sets were enriched for apoptosis and good differentiation. Although when possible the grade of the tumor was included in the Cox model, and therefore should not be a confounding variable, it is possible that histological grade does not completely account for the differentiation of a tumor, indicating the importance of genomics for accurate profiling.

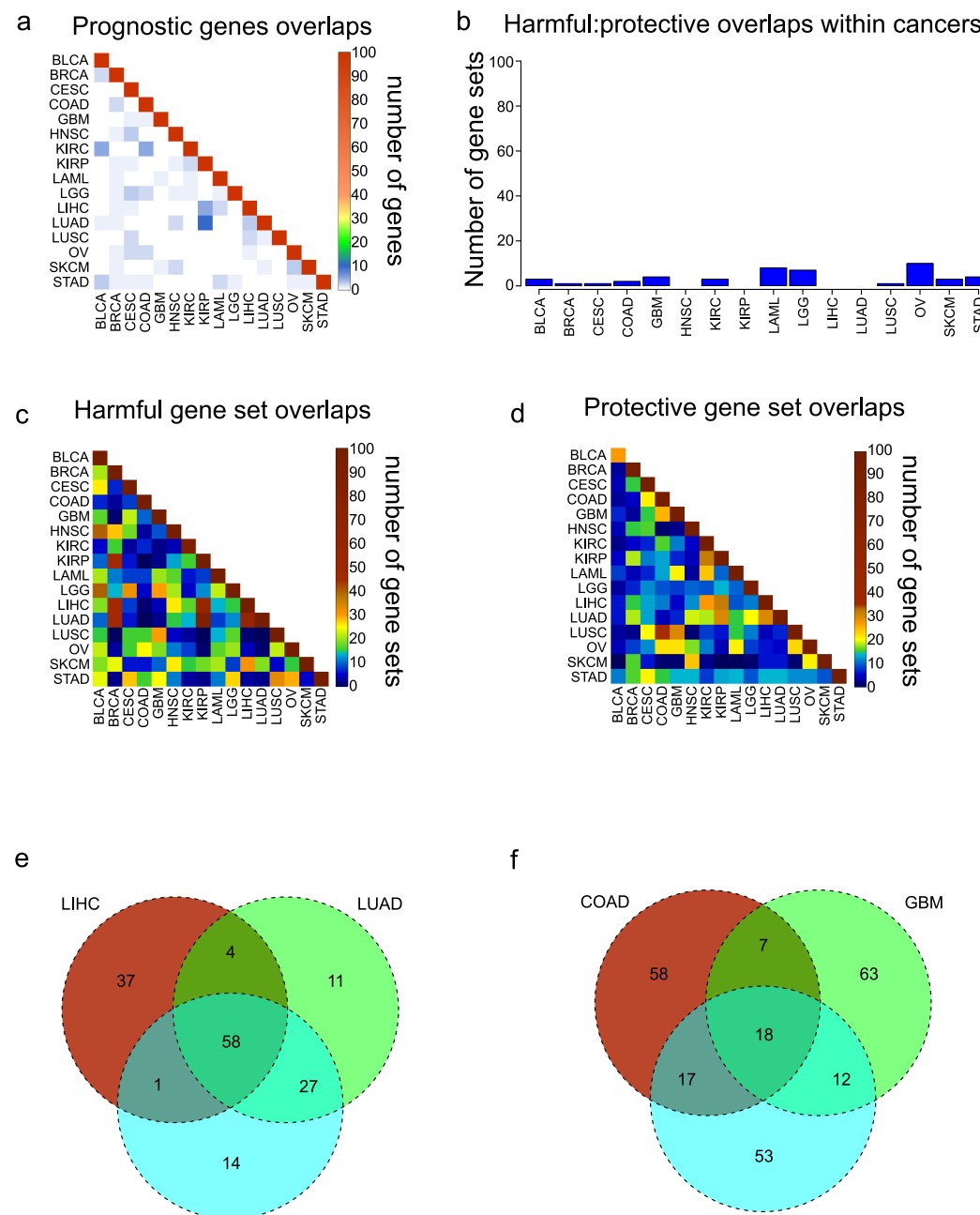

**Figure 2 Overlaps of prognostic genes and gene sets.** (A) Heatmap displaying the overlaps between cancers of the 100 most significant genes of each cancer. (B) Overlaps within cancers of the 100 most significantly enriched gene sets for protective genes, and the 100 most significantly enriched gene sets for harmful genes. (C, D) Overlaps between cancers of the 100 most significantly enriched gene sets for harmful genes (C) and protective genes (D). (E) Venn diagram showing the overlaps of the 100 harmful gene sets for LIHC, LUAD, and KIRP. (F) Venn diagram showing the overlaps of the 100 protective gene sets for COAD, GBM, and LUSC.

## Cancers can be clustered by gene and gene program Cox coefficients

To date, different cancers have been compared to each other through mRNA levels, miRNA levels, protein levels, networks, copy number alterations, DNA methylation, somatic mutations or some combination of these (*Akbani et al., 2014*; *Ciriello et al., 2013*; *Hamilton et al., 2013*; *Hoadley et al., 2014*; *Kandoth et al., 2013*; *Knaack, Siahpirani & Roy, 2014*). The Cox coefficients in our analysis contain a level of information not present in any of these data types, and consequently can potentially reveal similarities or differences between cancers that were not appreciated before. Therefore we sought to attempt to cluster cancers using Cox coefficients of genes instead of expression levels. Because the Cox models for the different cancers contain different numbers of covariates, and different strengths of gene expression correlation to survival, the range of values of the Cox coefficients vary between cancers. To correct for this, we normalized the coefficients for each cancer using a sigmoidal function which robustly scaled both negative and positive coefficients to their 95th percentile values (see methods). In addition, whereas every gene has an expression value, only significant prognostic genes have Cox coefficients appreciably above or below 0. Performing clustering with large numbers of nonsignificant genes which all have very similar values for every cancer will only add noise to the clustering. As a result, we limited our clustering to genes which had a FDR less than or equal to .05 in at least four of the sixteen cancers.

Hierarchical clustering of the 16 cancers was performed with the sigmoidal normalized Cox coefficients of this set of genes (Fig. 3). The clustering grouped LIHC, LUAD, and KIRP together, which were the same cancers that shared the highest number of harmful gene sets. In addition, GBM, COAD and LUSC clustered together, which were the cancers that had the highest number of protective gene sets overlap. The fact that two separate methods, using different sets of genes, were able to find these groupings of cancers gives us confidence that this finding is robust and biologically significant.

We next wanted to know if there were established pathways that distinguished our groupings of cancers from each other. Using a list of nonredundant gene programs that have been shown to distinguish cancers from one another on the basis of expression levels (*Hoadley et al., 2014*), we sought to distinguish cancers using Cox coefficients of pathways. For each pathway the average sigmoidal normalized Cox coefficient was calculated in each cancer. Because a Cox coefficient can be positive or negative, if a pathway has some genes which are protective and some genes which are harmful, the average Cox score will be near zero. In addition, if a pathway only contains genes which are not prognostic, all of those Cox scores will be near zero and the pathway score will be near 0. The only way for a pathway to have a positive or negative score is for it to contain prognostic genes which are either consistently protective or consistently harmful.

Using the Cox scores for these 22 gene programs, we again performed hierarchical clustering (Fig. 4). We column scaled the values to highlight which gene programs are most important for each cancer. Overall the same groupings we had seen with gene sets and individual genes were recapitulated, with LUAD, KIRP, and LIHC again forming a cluster

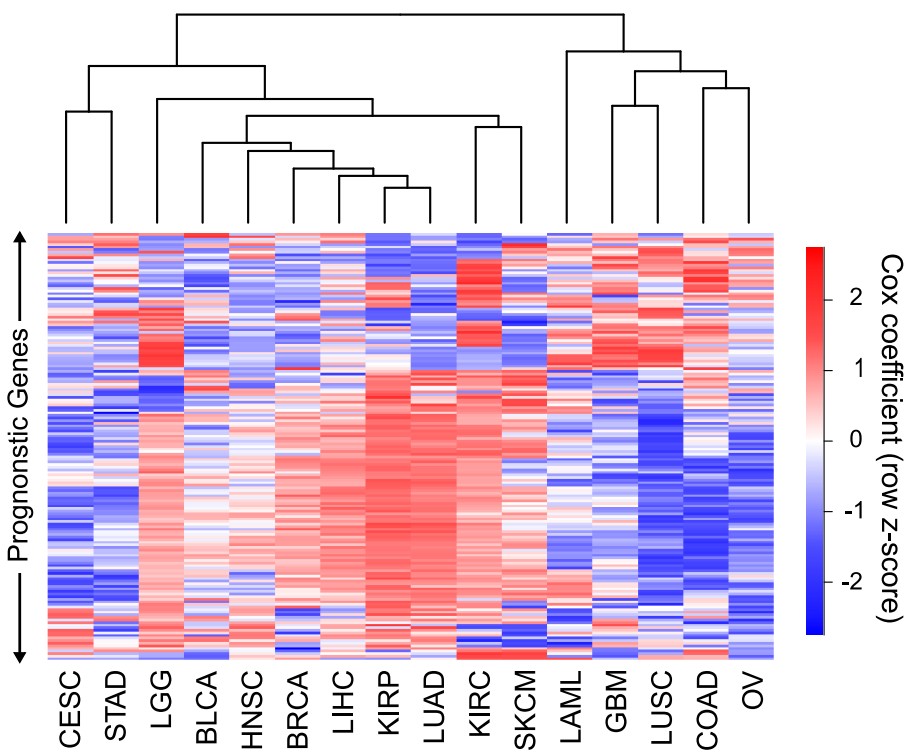

**Figure 3 Clustering of cancers using gene Cox coefficients.** Clustering of genes and cancers using the sigmoidal normalized Cox coefficients of a list of genes that had an FDR less than or equal to .05 for at least four cancers. Pearson correlation distance metric was used for both row and column clustering, and Cox coefficients were row scaled (*z*-score).

and COAD, LUSC, and GBM grouped together. In the LUAD/KIRP/LIHC group poor prognosis is associated with high proliferation rates and glycolysis, while good prognosis is associated with apoptosis and a dependency on oxidative phosphorylation. In contrast, for GBM/LUSC/COAD, proliferation is protective while EGF response predicts poor survival.

We also found cancer specific protective/harmful pathway enrichments that are consistent with known cancer biology. For example, in KIRC the highest intensity gene program is "fatty acid oxidation," and KIRC is a cancer that is known to depend on dysregulation of metabolism and is a classic example of the "Warburg effect" (*Linehan, Srinivasan & Schmidt, 2010*). Our results show that patients with high expression of genes utilizing oxygen survive longer, which underscores the importance of a metabolic shift in this cancer. As another example, EGFR is the most commonly mutated gene in GBM (*Brennan et al., 2013*), and we observed that increased EGFR activity is associated with poorer outcomes. And BLCA and SKCM, which are known for being responsive to immunotherapy, both benefit from increased interferon response and an immune cell signature which is likely a proxy for immune cell infiltration.

## DISCUSSION

Cancer researchers are increasingly looking to focus on factors which have clinical significance, and many different resources now allow researchers to identify if a protein

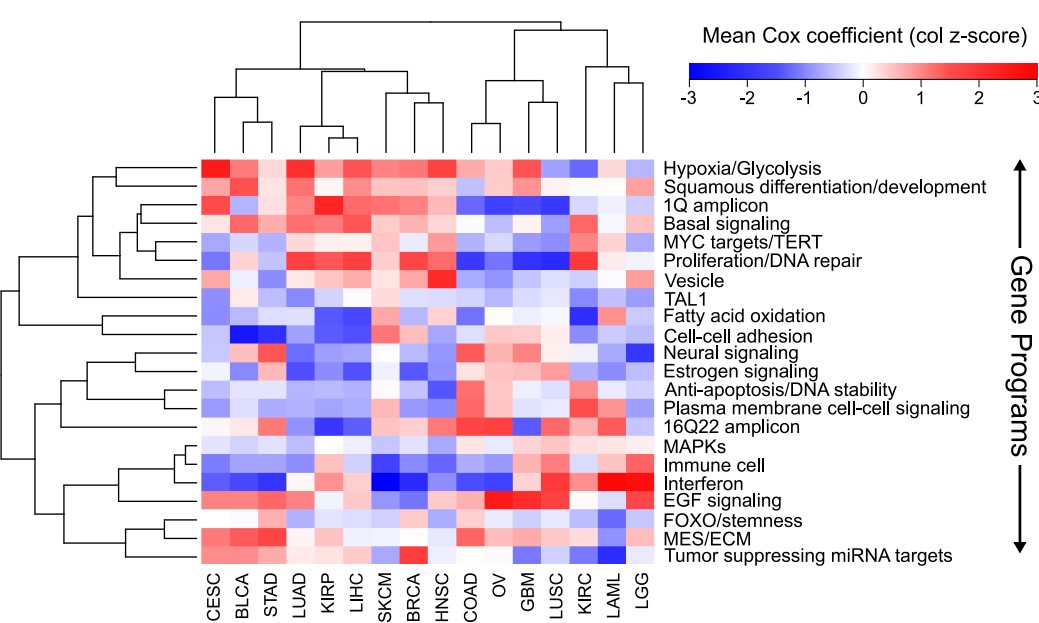

**Figure 4 Clustering of cancers using gene programs.** Using an established list of gene programs (see methods and Table S3), cancers and gene programs were clustered using the means of sigmoidal normalized Cox coefficients of the genes present in each program. Pearson correlation distance metric was used for both row and column clustering, and the average Cox coefficients were column scaled (z-score).

of interest has clinical implications, including OMIM, dbSNP, ClinVar, cBioPortal, FINDbase, and others (*Hamosh et al., 2005*; *Landrum et al., 2014*; *Papadopoulos et al., 2014*; *Smigielski et al., 2000*). Despite this, it currently is not possible to find comprehensive lists of genes which are associated with survival in different cancers. Using recently available RNA-SEQ and clinical data from the TCGA for 6,495 patients, we correlated every expressed annotated gene to survival in 16 different cancers, providing the scientific community with thousands of highly significant genes for further study.

There is an unexpectedly large variation between cancers in the number of statistically significant prognostic genes, which should be used to inform our evaluation of prognostic genes from different cancers. For example, a significant *p*-value for a gene from a cancer such as LGG or KIRC should not be surprising, given the thousands of genes that survive a stringent *p*-value cutoff in these tumors (Table 1, Fig. 1A and Fig. S1). In contrast, weaker *p*-values for predicting prognosis in cancers such as STAD or COAD are still biologically important although they have no genes that pass a stringent *p*-value threshold for biological significance (Table 1, Figs. 1A and 1C).

RNA-SEQ is a relatively new technology, and its ability to identify prognostic genes in many cancers has not been explored. Although the number of expressionally prognostic genes (EPGs) varied among cancers, regardless of the cancer we identified expression profiles which significantly separated patients into high risk and low risk groups. One of the main advantages of RNA-SEQ over microarrays is the ability to identify unannotated transcripts. In fact, recent studies have investigated the expressions of pseudogenes and

long noncoding RNAs in large numbers of TCGA RNA-SEQ data sets (*Han et al., 2014*; *Iyer et al., 2015*). It would be interesting to see if these transcripts show the same trends as protein coding genes across these cancers.

This comprehensive analysis of prognostic genes allowed us to explore the ability of the prognostic genes themselves, enriched gene sets, and Cox coefficients (a measure of strength of correlation to better or worse survival) to identify similarities and differences among cancers. We found that the most prognostically significant genes were not shared between cancers. However, we found that protective genes and harmful genes are enriched in very different gene sets, and there were large overlaps of these gene sets for LUAD, LIHC, and KIRP, and for COAD, LUSC, and GBM. We were able to recapitulate these findings by clustering with both Cox coefficients of individual genes, and average Cox coefficients of gene programs, suggesting that these findings are biologically significant and that we have identified a paradigm for incorporating genomic and clinical data to compare cancers.

Although it is important not to mistake a correlation for causation, our analysis suggests intriguing insights into the pathogenesis of different cancers. For example, currently EGFR inhibitors are recommended for LUAD patients with EGFR mutations, but EGFR mutations are rare in LUSC and patients with mutations do not respond well to tyrosine kinase inhibitors (*Chiu et al., 2014*). Despite this, response rates to EGFR inhibitors for LUSC studies are threefold higher than expected (*Chiu et al., 2014*), suggesting that although EGFR itself may not be mutated, responders may still have a cancer which is dependent on EGFR signaling. This is consistent with our gene program analysis, where EGFR response was most strongly associated with poor survival in LUSC, and LUSC was consistently associated with GBM, which is a cancer known for EGFR dysregulation. This suggests that using a measure of EGFR activity other than mutational status could be used to find LUSC patients that would benefit from a tyrosine kinase inhibitor. In addition, our analysis may be used to suggest treatments for cancers which are not well studied. For example, KIRP does not have successful treatments and there is a current search for drugs which may be of benefit (*Schuller et al., 2015*). Our analysis suggests that the pathogenesis of KIRP is very similar to LIHC and LUAD, indicating that treatments currently used for those cancers may be able to be co-opted for KIRP.

This analysis is among the first attempts at using clinical correlations to compare cancers. Although we utilized the most up to date information possible, well established statistical techniques, and obtained robust findings, there are many ways this type of analysis can be improved. For example, it is now being recognized that cancer is not a single disease, but rather a group of molecularly and clinically distinct diseases which share a tissue of origin. Through a combination of genomic measurements, the TCGA Research Network has divided individual cancers into four or five subtypes, for example GBM has been divided into proneural, neural, classical, and mesenchymal subtypes (*Brennan et al., 2013*). Currently, clear subtypes have not been found for all 16 of the cancers in this study, and for many cancers dividing the cancers into the subtypes would result in a loss of power due to the limited number of patients. However, as these classifications are refined and the number of patient samples continues to grow, a natural extension of this study would be to repeat

it for individual subtypes, which would potentially decrease the heterogeneity of the data. In addition, treatment is one the largest confounding variables in survival analyses, but the TCGA pharmacological data is currently incomplete making it impossible to incorporate this information into the model. Despite these current limitations, this study has shown that incorporating clinical information into pan-cancer analyses is capable of yielding insights into cancer pathogenesis that have thus far been unappreciated by other methods.

## ACKNOWLEDGEMENTS

We acknowledge the contributions of the TCGA Research Network and its TCGA Pan-Cancer Analysis Working Group led by JM Stuart, C Sander and I Shmulevich. Without their efforts this type of analysis would not be possible.

### Funding

JA was supported by the Cell and Molecular Biology Training Grant T32-GM008136 and by an F31-CA189446 from the NIH. The work was supported by R01-CA60499 to AD from the NIH.

### Grant Disclosures

The following grant information was disclosed by the authors:
Cell and Molecular Biology Training Grant: T32-GM008136.
NIH: F31-CA189446, R01-CA60499.

### Competing Interests

Jordan Anaya has started a company, Omnes Res, that may carry out analyses similar to that reported in this paper. The other authors have no competing interests to declare.

### Author Contributions

- Jordan Anaya conceived and designed the experiments, performed the experiments, analyzed the data, wrote the manuscript and prepared the figures and/or tables.
- Brian Reon conceived and designed the experiments.
- Wei-Min Chen contributed reagents/materials/analysis tools.
- Stefan Bekiranov contributed reagents/materials/analysis tools and reviewed drafts of the paper.
- Anindya Dutta conceived and designed the experiments, contributed reagents/materials/analysis tools and reviewed drafts of the paper.

### Data Availability

GitHub: https://github.com/OmnesRes/pan_cancer.

## Supplemental Information

Supplemental information for this article can be found online at http://dx.doi.org/10.7717/peerj.1499#supplemental-information.

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
