# Peer review of "A pan-cancer analysis of prognostic genes"

_PeerJ, doi:10.7717/peerj.1499_

## Round 0.1 · original submission · Major Revisions

We would be grateful if you could address the reviewers comments in a revised manuscript and provide a point-by-point response to the concerns.

Reviewer 1 ·

Basic reporting

This article presents a very interesting analysis using mRNA-Seq data from TCGA to identify gene expression signatures that have prognostic significance. The article identified significant deleterious and protective genes that were shared and mutually exclusive across different cancer types. Clustering, p-values of genes correlated with prognosis, identified cancer types with similar and dissimilar overall survival. The article also reports cellular mechanisms significantly correlated with the disease outcomes using gene set enrichment analysis.

However, several questions regarding the analysis and the conclusions need to be addressed

Experimental design

First, the title suggests that a pan-cancer approach to identifying prognostic genes is used, however the Cox coefficient were calculated at a cancer type level. The advantage of pan-cancer analysis is the additional power that is drawn from pooling samples from multiple cancer types. This has not been fully utilized in the study. Given this, how is this body of research adding to or different from previous reports by Zhao et. al. 2015, Yang et. al. 2014 and Zhang et. al. 2014.

Secondly, multiple gene expression based prognostic indices for most cancer types have been developed and are being used in the clinic. It would be interesting to see how, prognosis identified by the method in the paper compares to previously developed methods. Moreover, it is well known that certain sub-types of cancers have a clear poor and good outcomes. It is understandable that clustering of Cox coefficients may not be feasible, given the sample size, for most cancer types. However, some cancer types such as BRCA which having over 900 samples should have sufficient power to readily distinguish Luminal A samples which have good overall survival compared to Basal-like samples which have poor outcomes. Establishing the proof of principal through replication of well established concepts is crucial when developing novel methods and this has not been done. Additionally, replicating the results reported in this article, ideally using independent data-sets, or by splitting the data into test and validation cohorts will address questions regarding reproducibility of these findings.


Lastly, correlation of survival may not correlate with causation or progression of the disease. Moreover, multiple genetic mechanisms contribute to parthenogenesis in cancer which may or may not lead to expression changes in the tissue. However, a substantial portion of the article delves into gene set enrichment which may lead to incoherent conclusions. For example, on line 360, the article suggests that KIRP cancers can benefit from treatment modalities used in LIHC and LUAD. Such sweeping generalizations should be made with extreme caution especially when no in vitro and in vivo validation is done.

Validity of the findings

No comments

Additional comments

No Comments

·

Basic reporting

No comments

Experimental design

No comments

Validity of the findings

No comments

Additional comments

Author of the paper “A pan-cancer analysis of prognostic gene”, showed the effectiveness of gene clustering into two groups based on the cox-coefficient for the classification of cancers using RNA-seq count data. This paper supports their claim with rigorous statistical analysis and meaningful biological interpretation. It is acceptable for publication with minor modification as below:
1. Line # 31: “we have investigated”; This paper is written by single author therefor, “I have investigated”.
2. Line # 131 and 132: Briefly define and describe RSEM and RPKM for better readability.

Reviewer 3 ·

Basic reporting

The author made claims that TCGA has not provided prognostic gene lists, but some studies have provided survival analyses by mutated genes. I would recommend the author check out papers such as TCGA's GBM analysis to ensure their claim does not conflict with such papers. The author should remove subjective and exaggerated language used throughout the Results section, eg. "...LGG has a 50% chance..." when its closer to 43%, "...BRCA has by far the largest...", etc. Some descriptions need clarification such as "..average number of expressionally prognostic genes (EPGs).", it is not clear from what the average is calculated. Additional clarity is needed in Table 1 to describe "Events", to describe the age at diagnosis (is it the average age of patients sampled?), to and why the genes in each cancer type is varied. It should be clear from the table column headings what exactly is being shown and the units of values (Median Survival (days)). Figure 1 has six panels, but only three sub figure labels. It is not clear how the left column of figures differs from the right column.

Experimental design

I wonder why the author only used 100 genes when 7000+ are available in figure 1 clustering. I can understand computational limitations may be a factor in avoiding all ~7k genes, but does the analysis break down when using increasingly larger sets?

Validity of the findings

With large discrepancies in the obtained results (table1), few patterns seemed to emerge. I'm not entirely convinced of any significant patterns the author claims to have found, but it seems as though the author recognizes this and dealt with it. Downstream analyses appear reasonable.

---

## Round 0.2 · accepted · Accept

Can you Incorporate reviewers 1 suggestions with Figure 1b edits along with the final version at Production stage.

Reviewer 1 ·

Basic reporting

The author has addressed most of my concerns.

I do believe " An analysis of prognostic genes across 16 cancer types" is a more appropriate title for the paper.

The correlation plot showing high correlations of Cox-coeffecients of the full dataset to its subset addresses concerns about reproducibilty of the result.

Experimental design

I can see how with subtype specific analysis using all cancer types may not be possible. I would suggest figure 1b (heatmap of inverse normal transformed expression values ) to be extended to include a legend (color side bar) indicating the sub-type of the patients. While we expect protective and harmful genes are sub-type specific, we might see trends contradictory. This would present a comprehensive picture to the readers of the utility of the methodology. I would also like to see similar analysis for an additional cancer-type, say BRCA, where sub-types have very distinct prognosis.

Validity of the findings

The findings are coherent and well whetted.

·

Basic reporting

Very clear and coherent

Experimental design

"No comments"

Validity of the findings

It is significant for the understanding of cancer biology as shown by robust statistical analysis.

Additional comments

Author of the paper “A pan-cancer analysis of prognostic gene ”, has addressed all the major concern and incorporated needful changes with clarity for publication.

Reviewer 3 ·

Basic reporting

No Comments

Experimental design

No Comments

Validity of the findings

No Comments

Additional comments

My prior comments have been settled. Thank you.